# The Impact of Sea Ice Cover on Microbial Communities in Antarctic Shelf Sediments

**DOI:** 10.3390/microorganisms11061572

**Published:** 2023-06-14

**Authors:** Marwa Baloza, Susann Henkel, Sabine Kasten, Moritz Holtappels, Massimiliano Molari

**Affiliations:** 1Alfred Wegener Institute Helmholtz Centre for Polar and Marine Research, Am Handelshafen 12, 27570 Bremerhaven, Germany; susann.henkel@awi.de (S.H.); sabine.kasten@awi.de (S.K.); 2Faculty 2 Biology/Chemistry, University of Bremen, Leobener Str., 28359 Bremen, Germany; 3Faculty of Geosciences, University of Bremen, Klagenfurter Str., 28359 Bremen, Germany; 4MARUM—Center for Marine Environmental Sciences, University of Bremen, 28359 Bremen, Germany; 5HGF-MPG Joint Research Group for Deep-Sea Ecology and Technology, Max Planck Institute for Marine Microbiology, 28359 Bremen, Germany; mamolari@mpi-bremen.de

**Keywords:** benthic microbial communities, iron reducers, dissolved iron, pore-water profiles, redox zones, Sva1033

## Abstract

The area around the Antarctic Peninsula (AP) is facing rapid climatic and environmental changes, with so far unknown impacts on the benthic microbial communities of the continental shelves. In this study, we investigated the impact of contrasting sea ice cover on microbial community compositions in surface sediments from five stations along the eastern shelf of the AP using 16S ribosomal RNA (rRNA) gene sequencing. Redox conditions in sediments with long ice-free periods are characterized by a prevailing ferruginous zone, whereas a comparatively broad upper oxic zone is present at the heavily ice-covered station. Low ice cover stations were highly dominated by microbial communities of *Desulfobacterota* (mostly *Sva1033*, *Desulfobacteria*, and *Desulfobulbia*), *Myxococcota*, and *Sva0485*, whereas *Gammaproteobacteria*, *Alphaproteobacteria*, *Bacteroidota*, and *NB1-j* prevail at the heavy ice cover station. In the ferruginous zone, *Sva1033* was the dominant member of Desulfuromonadales for all stations and, along with eleven other taxa, showed significant positive correlations with dissolved Fe concentrations, suggesting a significant role in iron reduction or an ecological relationship with iron reducers. Our results indicate that sea ice cover and its effect on organic carbon fluxes are the major drivers for changes in benthic microbial communities, favoring potential iron reducers at stations with increased organic matter fluxes.

## 1. Introduction

Shelf sediments play a significant role in the remineralization of organic matter (OM) and the recycling of nutrients and trace metals, e.g., [1,2,3,4]. In shelf regions, photosynthetic primary production accounts for 30–40% of organic matter supplied to the sea floor [5]. The majority of OM deposited at the seabed is remineralized while part of it is buried permanently, e.g., [6,7]. The remineralization of OM is mainly driven through microbially-mediated metabolic reactions using sequences of terminal electron accepting processes (TEAPs) with electron acceptors such as O_2_ for aerobic respiration followed by NO_3_^−^, Mn(IV), Fe(III), SO_4_^2−^, and CO_2_ for anaerobic respiration, e.g., [3,8,9,10,11]. The microbial remineralization of OM through the TEAPs is the main driver for biogeochemical processes in marine sediments [12,13,14,15] and is responsible for the formation of a typical/distinct sedimentary redox zonation, e.g., [8,11,16]. The extent of the different redox zones and the depth position of redox boundaries are mainly controlled by the reactivity and availability of terminal electron acceptors and OM accumulation rates, e.g., [8,16,17,18], suggesting that each of these zones is shaped by microbes with specific metabolic traits [19,20]. Therefore, differences in microbial community composition have been successfully used to decipher the redox state and biogeochemical processes in a wide range of coastal marine and deep-sea sediments [21,22,23,24,25,26,27,28].

In the area of the Antarctic Peninsula (AP), the amount of organic matter originating from surface-water primary production is highly variable—both temporally and spatially—and depends mainly on the availability of light, which is regulated by season and sea ice cover [29,30,31]. During the spring-summer season, the melting of sea ice provides light for surface-water primary production with maximum production at the marginal ice zone [29]. On the other hand, the growth of sea ice cover during winter reduces light penetration, thereby limiting primary production. Sea ice cover thus controls the flux of OM to the seafloor. Baloza et al. [32] found that moderate sea ice cover (5–35%), which combines both favorable light conditions and water column stratification for algal growth, correlates with a high OM supply rate to the seabed. Previous studies have demonstrated that the quantity and quality of OM determine the microbial community composition, with certain taxonomic groups showing strong correlations with sedimentary parameters such as chlorophyll a concentration [23,33] and total organic carbon content [34,35,36].

Another factor regulating the production and subsequent deposition of OM over large parts of the Southern Ocean is the availability of dissolved iron (DFe), a limiting micronutrient for phytoplankton productivity, e.g., [37,38]. Iron supply can enhance phytoplankton growth resulting in higher sequestration of atmospheric CO_2_ and enhanced accumulation of phytodetritus on the sea floor, a process known as the biological carbon pump [39,40]. Iron enters the Southern Ocean from various sources, with considerable variability in their estimated contribution. This includes iceberg-rafted debris (IRD) to sub-ice shelf, which contributes about 180–1400 Gg a^−1^ of bioavailable Fe [41], followed by anoxic shelf sediments (7–790 Gg a^−1^) [3], aeolian input (1.12 Gg a^−1^) [41,42], and anoxic subglacial meltwaters (0.03–5.9 Gg a^−1^) [43]. The importance of anoxic shelf sediments as a potential source of dissolved iron to the water column has recently been reported for Antarctic shelf sediments [3,32,44,45], and this source appears to be at least as significant as the input of iceberg–hosted material [3,44,46,47,48]. Therefore, identifying microbial key players involved in the iron cycle is essential, especially for the Southern Ocean, to better understand the factors that control iron bioavailability and efflux into the water column. At the moment, however, baseline knowledge about iron-reducing microbial taxa in Antarctic sediment is very limited [26,49].

In shelf sediments, dissimilatory microbial iron reduction (DIR) plays a key role in the remineralization of organic matter [3,26,27,28,32,45,50,51]. Predominantly, the microbial communities identified in performing DIR include bacterial taxa belonging to the order *Desulfuromonadales* (mostly those of the genera *Desulfuromonas*, *Desulfuromusa*, *Pelobacter*, *Geopsychrobacter*, and *Geothermobacter*) [52,53,54,55,56,57]. Furthermore, *Sva1033*, a clade of *Desulfuromonadales*, was identified by Ravenschlag et al. [58] and Wunder et al. [26] using RNA stable isotope probing experiments with permanently cold marine sediments from the Arctic and Sub-Antarctic, respectively. Both studies showed that the clade *Sva1033* has iron-reducing capabilities.

Sedimentary redox conditions along the eastern shelf of the AP have recently been investigated by Baloza et al. [32], who found that the supply of OM to the seafloor is mainly controlled by a sea ice cover. In areas of heavy ice cover, the low carbon supply coincides with low rates of benthic carbon remineralization. There, the surface sediments were characterized by a comparatively broad upper oxic zone of up to 4 cm with an underlying comparatively narrow ferruginous zone. With decreasing sea ice cover, an increase in carbon supply and benthic carbon remineralization rates was observed. At these sites, high dissolved iron concentrations of up to 400 µM were found at very shallow depths and below the ferruginous zones. A gradual decrease in sulfate concentrations and a steep increase in H_2_S concentrations marked the beginning of the sulfidic zone [32].

The present work on the microbial communities is complementary to the geochemical findings of Baloza et al. [32] and is based on the same set of sediment samples. The study is intended as a base study to evaluate the impact of changing sea ice conditions on benthic microbial communities and—ultimately—nutrient fluxes across the sediment–water interface. We hypothesize that (i) the impacts of sea ice cover on organic matter fluxes are responsible for changes in benthic microbial communities, and (ii) the strong changes in redox zonation, induced by intense microbial organic matter remineralization rates, are coupled with an increment of iron-reducers at stations with increased organic matter fluxes. In order to test the two hypotheses, sediment samples from five stations along a 400 nautical mile transect with contrasting sea ice conditions were investigated by 16S ribosomal RNA (rRNA) gene sequencing. Correlation, multivariate regression, and differential taxa abundance analyses were performed to identify the main factors shaping the microbial community composition.

## 2. Materials and Methods

### 2.1. Sample Collection

During research cruise PS118 with the German research vessel RV POLARSTERN (February–April 2019), sediments were collected from five stations along a 400-nautical mile transect from the eastern Antarctic Peninsula to the West of the South Orkney Islands (Figure 1; Table 1). In this study, we used sediments collected from five shelf stations at water depths ranging between 329 and 455 m to allow cross-comparison of sediment properties and OM turnover rates independent from water depth. The two deep stations (St5 and St6) were excluded from this study. At each station, a total of nine sediment cores with intact sediments and overlying water were collected from multiple deployments of a multi-corer (Oktopus GmbH, Kiel, Germany). Immediately upon retrieval, sediment cores were transferred to the ship’s cool laboratory and placed in a water bath at 0 °C. Three cores were sliced on board to 0–1, 1–2, 2–3, 3–5, 5–7, and 14–16 cm layers and stored at −20 °C for DNA analysis. For pore-water and solid-phase analysis, six different cores were used and analyzed, as described previously by Baloza et al. [32].

**Table 1 microorganisms-11-01572-t001:** Description of investigated stations, including water depth, ice cover, moderate sea ice index, and bottom water temperature. The ice cover (30-year average of daily sea ice concentration) was calculated from historical satellite data of the Sea Ice Index, version 3 [59]. The moderate sea ice index is the relative occurrence of moderate ice cover (defined as 5–35% ice cover), weighted by the length of daylight (sunrise to sunset), as described by Baloza et al. [32]. Total organic carbon (TOC) from the upper five cm depth. The Total Carbon Remineralization Rate was calculated from diffusive fluxes of O_2_, NH_4_^+^, DFe, DMn, and S^2−^ using the equation from Baloza et al. [32]. The Total Carbon Supply Rate was calculated from the TOC removed in the upper layer due to aerobic C-remineralization added to the TOC accumulation rate.

	Longitude	Latitude	Date	Water Depth	Ice Cover	Mod. Sea Ice Index	Bottom Water Temperature	TOC	Total Carbon Remineralization Rate	Total Carbon Supply Rate
	°W	°S	DD/MM/YYYY	m	%		°C	wt%	mmol m^−2^ d^−1^	mmol m^−2^ d^−1^
Shelf St1	57.75	64.98	04/03/2019	428	81	0.038	−1.9	0.71	1.134	2.67
Shelf St2	55.90	63.97	11/03/2019	415	49	0.127	−1.5	1.02	3.108	6.78
Shelf St3	55.74	63.80	14/03/2019	455	47	0.084	−1.2	0.19	1.452	3.20
Shelf St4	54.33	63.05	17/03/2019	447	33	0.174	−0.9	1.26	7.396	12.97
Shelf St7	46.55	60.93	28/03/2019	329	28	0.092	−0.1	0.60	1.833	2.50

**Figure 1 microorganisms-11-01572-f001:**
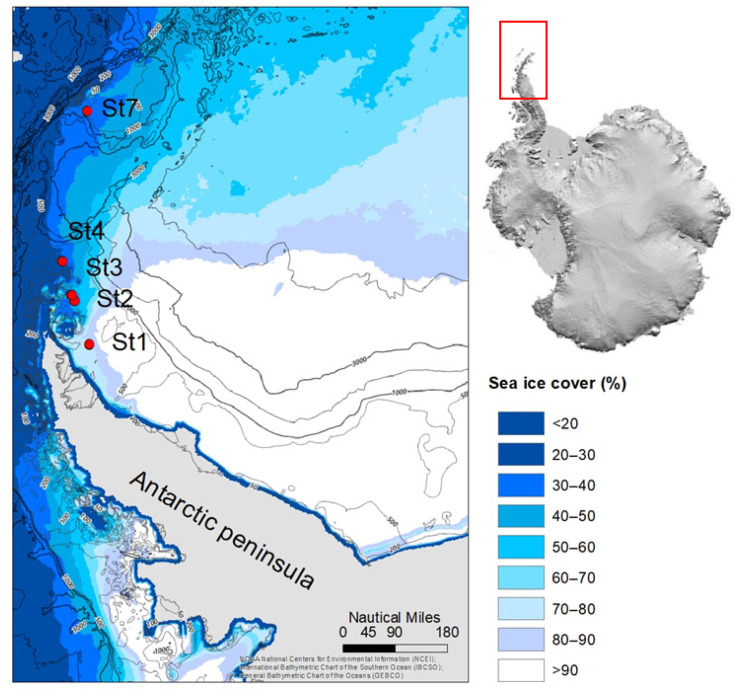
Map showing the Antarctic Peninsula with the locations of the sampled stations (red circles with station numbers) during the PS118 expedition modified after Baloza et al. [32]. Colors denote the proportion of time the ocean is covered by sea ice of concentration 85% or higher as calculated from AMSR-E satellite estimates of daily sea ice cover at 3.37 nautical mile resolution [60]. Water depth is indicated by isolines.

### 2.2. DNA Extraction and Sequencing

The DNA was extracted from 0.25 g of wet sediment using the DNeasy PowerSoil Kit (Q-BIOgene, Heidelberg, Germany) following the protocol provided by the manufacturer. Amplicon sequencing was conducted on an Illumina MiSeq machine at AWI laboratory. For the 16S rRNA gene amplicon library preparation, we used the bacterial primers 341F (5′-CCTACGGGNGGCWGCAG-3′) and 785R (50-GACTACHVGGGTATC TAATCC-30), which amplify the 16S rRNA gene hypervariable region V3–V4 in Bacteria (400–425 bp fragment length). The amplicon library was sequenced with MiSeq v3 chemistry in a 2 × 300 bp paired-end run with >50,000 reads per library, following the standard instructions of the 16S Metagenomic Sequencing Library Preparation protocol (Illumina, Inc., San Diego, CA, USA).

### 2.3. Amplicon Data Analysis

The quality cleaning of the sequences was performed using several software tools. The first tool was Cutadapt [61], which was used to remove the attached primers from the data. Then, DADA2 packages in R [62] were used for filtering and trimming low-quality sequences. Further, sequences with more than two expected errors were removed depending on their quality profiles. DADA2 was also used to merge the filtered forward and reverse reads and remove Chimeras from the denoised output. For the taxonomic classification, the SILVA 138 database [63] was used, and sequences with less than 90% of similarity with SILVA sequences were removed from the dataset. Absolute singletons (i.e., which refer to ASV sequence occurring only once in the full dataset) [64], contaminant sequences (as observed in the negative control), and unspecific sequences (i.e., unclassified sequences at the domain level, chloroplast, and mitochondrial sequences) were removed from amplicon datasets before the statistical analysis. The total number of reads per sample was summarized in an amplicon sequence variants (ASVs) table (Appendix A).

All statistical analyses were conducted using RStudio v4.0.2; R Core [65]. Sample data matrices and alpha diversity indices were generated using the R package ‘phyloseq’ v1.32.0 [66]. The rarefaction curves for each sample were generated based on 50 equally spaced rarefied sample sizes with 100 iterations. Non-metric multidimensional scaling (NMDS) analysis was conducted on Euclidean distance matrix based on ASV abundances that have undergone centered log-ratio (CLR) transformation using R package ‘vegan’ v2.5.7 [67] and ‘Composition’ v2.0.4 [68], respectively. Differences in bacterial communities between groups of stations, defined based on sea ice cover, were tested with analysis of similarities (ANOSIM) in the R package ‘vegan’ v2.5.7 [67]. Furthermore, the significance of the relationship between the frequency of moderate sea ice conditions, sediment depth, and community composition was tested using permanova tests (9999 unrestricted permutations; *p*-values < 0.05) with ‘vegan’ v2.5.7 [67]. Geochemical parameters that shape microbial community compositions were standardized (z-scoring) and tested for predictor variable collinearity, then investigated to determine the correlations between geochemical parameters and bacterial community compositions by distance-based redundancy analysis (dbRDA) in the R package ‘vegan’ v2.5.7 [67]. The significance of the relationship between explanatory variables and community composition was tested using Monte Carlo permutation tests (9999 unrestricted permutations; with *p*-value < 0.05).

In order to identify taxonomical associations with iron cycling of all shelf stations, the Metastats-test-based differential abundance analysis of taxa was performed using the ALDEx2 package v1.20.0 [69] at a significance threshold of <0.01 for the parametric test (glm.eBH) and non-parametric test (kw.ep). This analysis was done to examine the different bacterial taxa between the upper 3 cm of sediment of Shelf St4 dominated by ferruginous conditions against the upper 3 cm of Shelf St1 in which oxic respiration is the dominant process for OM degradation. Linear regression between centered log-ratio values of the ASV highly abundant in the ferruginous zone and dissolved iron (DFe) concentrations (in µmol/L) for the five shelf stations was performed using Pearson’s correlations at *p* < 0.05. Furthermore, the environment coverage for the most closely related sequences (>99% similarity) to ASV highly abundant in the ferruginous zone was identified with BLASTn (GeneBank nucleotide database, 11 March 2022) and reported for each ASV.

## 3. Results

### 3.1. Bacterial Diversity and Sequencing Data Normalization

Using Illumina 16S rRNA amplicon sequencing of the V3–V4 hypervariable region, we obtained a final dataset of 4,252,065 reads (amplicons) in 90 samples, which were assigned to 38,655 ASVs belonging to bacterial taxonomies of 64 phyla, 162 classes, 378 orders, 536 families, and 786 genera. Rarefaction profiles of 16S rRNA gene sequences reached a plateau for most of the samples (Appendix A). Sediments hosted from hundreds to thousands of ASVs in each station were investigated. The highest bacterial richness (measured as the number of ASVs per sample) was observed in the uppermost sediments from all stations. However, bacterial diversity (measured with the observed richness, Shannon index and the inverse Simpson index) decreased northward in the AP surface sediments and differed among sediment depths (Appendix A, Appendix A). Overall, the heavy ice-covered station (St1) had the highest species richness and diversity of all stations.

Sample read counts ranged from 942 to 250,824 per sample (Appendix A), the median sequencing depth was 30,167, and the 75th percentile and the 25th percentile were 55,749 and 18,651, respectively (Appendix A). To check whether the uneven sequencing depth could introduce bias in the analysis of microbial communities between stations/layers, we tested if there is a significant correlation between the original dataset and the dataset that is normalized by a fixed number of reads. We normalized the dataset to the median sequencing depth (30,000) and to the smallest number of reads (942). Our results showed high correlations between the original dataset and the dataset that is normalized to the median sequencing depth (Mantel statistic r: 0.99, Significance: 1 × 10^−4^). In contrast, the original dataset and the normalized dataset to the smallest number of reads did not significantly correlate (Mantel statistic r = 0.047, Significance: 0.2035). Based on this result, the original dataset was used for further analysis.

### 3.2. Effect of Sea Ice Cover and Geochemistry on Bacterial Community Structure

Non-metric multidimensional scaling of ASVs did not show any clear pattern between stations (Figure 2A). However, when similarity in bacterial community structure was investigated between the stations grouped according to sea ice cover, the ANOSIM showed that bacterial communities between heavy ice cover and low ice cover stations are significantly different (ANOSIM, r > 0.53, *p* < 0.001). This result suggests that sea ice cover is a variable affecting the microbial community composition in the AP shelf sediments. To further test the contribution of sea ice cover in explaining differences in the microbial community, a permanova test was carried out, also including sediment depth as an explanatory factor, which is an important environmental constraint for microbes (Appendix A). The permanova test showed that the frequency of moderate sea ice conditions (here defined as 5–35% sea ice cover) is significantly different between stations (*p* < 0.0001) and can explain 4% (when testing all sediment depths) to 13% (when testing the surface layers only) of the variation between microbial communities. Sediment depth was also tested as another factor affecting the microbial community structure. The results showed a significant difference between sediment depths (*p* < 0.0001), explaining 9% of the variance.

In order to identify the potential geochemical parameters that correlate with the microbial communities across all sites, a dbRDA was performed (Figure 2B, F = 2.90, *p* < 0.001, Df: 7, 81). The factors of O_2_, DFe, DIC, SO_4_^2−^, Fe(III)/Al, S/Al, and the C/N ratio were included as explanatory variables in the model and together explained 20% of the total variation in the bacterial community. The variance inflation factor between these variables was below five (VIF = 4.3), indicating independence between variables. The total organic carbon content (TOC) was not used as an explanatory variable due to collinearity to other variables such as DIC, SO_4_^2−^, Fe (III)/Al and S/Al. NH_4_^+^. H_2_S was also removed from this data analysis as it correlated with DIC and SO_4_ ^2−^, respectively. However, we used the sulfur content in the solid phase (S/Al) instead of H_2_S as an indication of sulfate reduction and accumulation of pyrite.

Community composition between high ice cover and low ice cover stations was described mainly by the increase of O_2_ (F = 3.754, *p* = 0.003) and DFe concentrations (F = 3.703, *p* = 0.003), followed by S/Al (F = 2.190, *p* = 0.005), Fe/Al (F = 2.009, *p* = 0.005), then C/N ratio (F = 1.868, *p* = 0.011), DIC (F = 1.636, *p* = 0.025), and SO_4_^2−^ (F = 0.954, *p* = 0.476). In accordance with the NMDS ordination (Figure 2A,B), oxic layers at the heavy ice cover station exhibited a strong separation of bacterial communities from the low ice cover stations. However, anoxic layers at the heavy ice cover station tend to cluster together with the low ice cover stations. The model strongly attributed this distinction to O_2_ and DFe differences between groups.

### 3.3. Patterns in Bacterial Community Composition

Figure 3 shows the relative abundance of the 30 most abundant taxa (> 0.1%) across all five shelf stations. Across all stations, similarities in the distribution of benthic microbial communities have been observed. Dominant taxa belonged mostly to Gammaproteobacteria and Woeseia, Bacteroidota (mostly Bacteroidia), and NB1-j were present in all sediment depths and represented >15% of the total microbial community (Figure 3). Distinct differences in the benthic microbial communities between heavy ice cover (Shelf St1) and low ice cover stations (Shelf St2, St3, St4 and St7) were also evident. Surface sediments of the low ice cover stations were dominated by anaerobic bacterial groups belonging to Proteobacteria (i.e., Halioglobus, order_B2M28, OM60(NOR5) clade, family_Rhodobacteraceae, and order_BD7-8) (>13%), Desulfobacterota (i.e., order_Sva1033, family_Desulfocapsaceae, order_Desulfobulbales, phylum_Desulfobacterota, family_Desulfosarcinaceae, Sva0081 sediment group) (>16%) Sva0485 (3%), and Bacteroidota (i.e., Lutibacter, Lutimonas, family_Bacteroidetes BD2-2, class_Bacteroidia, family_Flavobacteriaceae) (10%) within the upper 3 cm of sediment. However, unlike the low ice cover stations, abundances of Desulfobacterota (order_Desulfobulbales, phylum_Desulfobacterota, family_Desulfosarcinaceae, Sva0081 sediment group) Sva0485, and Bacteroidota (i.e., Lutibacter, Lutimonas, family_Bacteroidetes BD2-2) were very low (<0.1%) in the top 3 cm of the heavy ice cover station (Shelf St1) where oxic respiration dominated OM degradation. At the same site, the relative abundance of Proteobacteria (i.e., order_BD7-8, class_Alphaproteobacteria, order_AT-s2-59) (>5%), Desulfobacterota (i.e., order_Sva1033, family_Desulfocapsaceae, phylum_Desulfobacterota) (>7%), Sva0485 (1.4%) and Bacteroidota (i.e., Lutibacter, Lutimonas, family_Bacteroidetes BD2-2, class_Bacteroidia) (8.5%) increased with depth (in the ferruginous zone). Importantly, Sva1033, a clade of Desulfuromonadia, was highly abundant within the top 3 cm of the sediment at the low ice cover stations (>7%) and at the sulfate reduction zones at 14 cm depth (>4%). Except for the high ice cover station, the highest abundance of Sva1033 occurred below 3 cm sediment at the ferruginous zone (>4%). In addition, sequences related to known sulfate reducers, such as Desulfobacterota (mostly Desulfobacteria and Desulfobulbia), Myxococcota (mostly Polyangia), and Sva0485 were highly abundant at 14 cm layer at low ice cover stations (14%) compare to the heavy ice cover station (<3%; at 14 cm layer).

### 3.4. Potential Iron-Reducing Bacteria

The results of the differential abundance analysis between the ferruginous zone at Shelf St4 against the oxic zone at Shelf St1 revealed that 791 ASVs were significantly different between the two stations (Shelf St1 + Shelf St4) (*p* < 0.05; Appendix A). Among these taxa, 12 ASVs contributed individually to more than 1% of the total community at the ferruginous zone, contributing together up to 14–20% at low ice cover stations in contrast to only <6% at the high ice cover station of bacteria community in the ferruginous zone. The major groups of bacteria present were identified as members of Sva1033 (class Desulfuromonadia), SEEP-SRB4 (class Desulfobulbia), Persicirhabdus (class Verrucomicrobiae), Halioglobus, order_B2M28 and OM60 (NOR5) clade (class Gammaproteobacteria), Maribacter and Lutimonas (class Bacteroidia) and class Polyangia.

Across all stations, the relative abundance of 12 putative Fe-reducing bacterial taxa show similar patterns along sediment profiles mirroring dissolved iron profiles (Figure 4). Sva1033 was the most abundant taxon in the ferruginous sediments, with a peak in the relative abundance at the same depth as the highest dissolved Fe concentrations (Figure 4). The relative abundance of Sva1033 was highest (>5%) near the sediment surface at low ice cover stations (Shelf St2, St3, St4, and St7) compared to the high ice cover station where the iron reduction zone is deeper (below 4 cm depth). There, its relative abundance was <2% at the surface and 0.3% below 5 cm depth (Figure 4).

Furthermore, significant positive correlations between centered log-ratio values of putative Fe-reducing bacteria and dissolved Fe concentrations were observed for all potential iron-reducers (*p* < 0.05), except for order_B2M28, where *p* = 0.08 (Figure 5). The class_Gammaproteobacteria (asv4) showed the highest correlation with DFe concentrations, followed by Lutimonas (asv26), while order_B2M28 (asv6) had the weakest correlation (Figure 5).

For the 12 putative Fe-reducing bacterial taxa, sequence homology searches within GenBank with the highest similarity (e.g., 100% or >99%) were performed using BLASTn (NCBI website) (Appendix A). The most closely related sequences were retrieved from the permanently cold marine sediment (e.g., Arctic and Antarctic; 31%) [58,70,71,72,73,74,75,76], from suboxic sediment (23%) [52,77], from methane seep sediment (17%) [77,78,79,80,81], from tidal flat sediment (17%), and from hydrothermal or mud volcanos sediments and deposits (9%) (Figure 6 and Appendix A).

## 4. Discussion

The Antarctic Peninsula is projected to undergo profound climatic and environmental changes affecting seasonal sea ice cover, water column stratification, terrestrial meltwater run-off, and related nutrient input, and thus the conditions for primary production, organic carbon (OC) export, and benthic remineralization [32,82,83]. The impact of sea ice cover on microbial communities in underlying sediments is currently understudied. We hypothesize that (i) the impact of sea ice cover on organic matter fluxes are responsible for changes in benthic microbial communities, (ii) the strong changes in redox zonation, induced by intense microbial organic matter remineralization rates, are coupled with an increment of iron-reducers at stations with increased organic matter fluxes. In order to test the two hypotheses, microbial community composition at different sites with contrasting sea ice conditions along the eastern shelf of the AP was analyzed using 16S ribosomal RNA (rRNA) gene sequencing in surface (top 16 cm) sediments. Our findings, as discussed below, confirmed our hypotheses, showing that the effect of sea ice cover on organic carbon fluxes is the major driver for the variability in benthic microbial communities and redox zonation in the sediments of the eastern coast of the AP. Importantly, at low ice cover stations, characterized by high carbon supply and remineralization rates, we identified the dominance of potential iron-reducing bacteria (up to 20% of total sequences; Figure 4).

In this study, we observed that benthic microbial communities are significantly different between sites with high and low ice cover. Recently, a biogeochemical study of sediments from the exact same stations [32] described sea ice conditions as the main factor controlling rates of organic carbon supply and benthic remineralization. A low carbon supply rate of 2.7 mmol C m^−2^ d^−1^ was measured for the station with heavy ice cover, explained by limited light availability and thus low surface-water primary production. In contrast, the locations with low sea ice cover showed high carbon supply rates of up to 13.0 mmol C m^−2^ d^−1^, indicating that primary and export production are both enhanced when both light and water column stratification is sufficient to support phytoplankton growth. This scenario is supported by known large phytoplankton blooms and high primary production during spring/summer seasons on the marginal ice zone along the AP [84,85,86,87]. Our results are in agreement with the previous studies on bacterial communities in marine Antarctic surface sediments [33,34]. Currier et al. [33] also found that the structure and composition of benthic microbial communities reflect the condition of sea ice cover. Microbial communities underlying first-year sea ice cover are highly dominated by heterotrophic algal polysaccharide degrading taxa and sulfate-reducing bacteria and correlate with higher chlorophyll a and total organic carbon content, reflecting increased surface-water productivity and high OC fluxes to the seafloor [33]. Conversely, in sediments underlying multi-year sea ice cover, an enrichment of known archaeal and bacterial chemoautotrophs was found with considerably lower chlorophyll a and TOC contents, reflecting low surface-water primary production [33]. In addition, Learman et al. [34] found that the variability in the benthic microbial community composition along the Antarctic surface sediment is mainly driven by the quality and quantity of organic matter and the availability of nutrients. These findings agree with the results of our study that the impact of sea ice cover on organic matter fluxes is the main factor structuring the benthic microbial communities along the AP in such a way that they select the microorganisms that best respond to the given conditions.

The gradual decrease in sea ice cover along the transect of the investigated sites, accompanied by an increase in organic C-supply rates, have resulted in a distinctly different redox zonation of the underlying sediments suggesting that the different redox conditions are shaped by organic C burial rates and the activity of microorganisms with specific metabolic traits. Sediments at the heavy ice cover station were characterized by a low carbon remineralization rate of 1.1 mmol m^−2^ d^−1^ and a deeper oxygen penetration depth compared to stations with low ice cover (6.3 ± 0.7 cm and 1.8 ± 0.02 to 0.5 ± 0.07 cm, respectively) (Appendix A), resulting in the upper boundary of the ferruginous zone to be located below 5 cm depth [32]. However, unlike the heavy ice cover station, sediments of the low ice cover stations showed high benthic carbon remineralization rates (1.8–7.3 mmol m^−2^ d^−1^), resulting in a more condensed redox zonation and high concentrations of dissolved Fe (>400 μM at Shelf St4) close to the sediment surface (Appendix A). Below the ferruginous zone, the gradual increase of H_2_S concentrations below 10 cm depth was detected, marking the beginning of the sulfate reduction zone (Appendix A). In accordance with these observations, we found that the differences in microbial community composition between the high ice cover and the low ice cover stations are largely explained by dissolved O_2_ and Fe concentrations in pore water (Figure 2B), which are fundamentally different in the uppermost sediment between stations with high and low ice cover (Figure 2B). This indicates that microbial communities in the surface sediments of the high ice cover station are mainly dominated by aerobes. In contrast, the variation in community composition over sediment depth at the stations with low ice cover was explained mainly by the variability of dissolved Fe concentration, S/Al, DIC, and SO_4_^2−^ (Figure 2B), suggesting that iron and sulfate-reducing bacteria are the dominant players.

Microbial communities in surface sediments (i.e., 0–1 cm) of all stations are dominated by bacterial groups belonging to *Gammaproteobacteria*, *Alphaproteobacteria*, *Bacteroidota*, and *NB1-j*, which were often detected during the initial degradation of algal-derived organic matter [22,75,88,89]. At the low ice cover stations, anaerobic bacterial communities of iron and sulfate reducers (31% at Shelf St4 to 41% at Shelf St7) were more abundant at these sites compared to the heavy ice cover station. The relative abundance of *Desulfobacterota* (mostly order_Sva1033, family_*Desulfocapsaceae*, phylum_*Desulfobacterota*), which harbor many species of iron reducer taxa [25,26,54,90] and sequences related to known sulfate reducers, such as *Desulfobacterota* (mostly *Desulfobacteria* and *Desulfobulbia*), *Myxococcota* (mostly *Polyangia*), and *Sva0485* were abundant near the sediment surface of low ice cover (>20%) and at greater depth (14 cm depth) (>12%). Similarly, some of those taxa have been previously observed in shallower depths in more reducing shelf sediments of the Sub-Antarctic island of South Georgia [26] and Arctic fjords [91]. Unlike the low ice cover stations, the relative abundance of anaerobic microbial communities of Desulfobacterota at the heavy ice cover station increased below the oxic zone and reached the highest relative abundance (>10%) within the ferruginous zone, reflecting low burial rates of organic matter.

Furthermore, the high relative abundance of sulfate reducers in the ferruginous zone at the sites with low ice cover can be explained by the tight link between the biogeochemical cycles of iron and sulfur. For this reason, Fe liberation into the pore water can result either from dissimilatory iron reduction (DIR) or happen abiotically related to sulfide oxidation by Fe(III) reduction, e.g., [92]. The tight coupling of the post-depositional iron and sulfur cycles has been demonstrated recently based on geochemical data from the Argentina continental margin [93], Arctic sediments [94,95] and (sub-)Antarctic sediments [26]. Wunder et al. [26] suggested that the high relative abundance of sulfate reducers in the ferruginous zones is due to sulfate reduction masked by the reoxidation of the produced sulfide to sulfate via Fe(III) reduction supporting the persistence of sulfate reducers.

The anoxic shelf sediments have recently been considered a potential source of bioavailable Fe to Antarctic coastal waters and beyond [3]. Along the sampling transect of the AP, elevated upward fluxes of DFe in the sediment were detected at the low ice cover stations (52 to 171 µmol m^−2^ d^−1^), while a low upward flux of DFe was observed at the heavy ice cover station (17.7 µmol m^−2^ d^−1^). The steep concentration gradients of DFe close to the sediment surface indicate that more DFe might escape from shelf sediments into the water column highlighting the importance of sediments underlying low ice cover as a potential source for limiting nutrients to the shelf waters [32]. To identify taxa potentially involved in the sedimentary biogeochemical cycling of iron, we investigated differences in bacterial community composition under different DFe fluxes.

At all sites, the distribution of *Sva1033*, a clade of Desulfuromonadia, is tightly coupled to the increase in the dissolved iron concentration showing the steepest slope in the linear regression (Figure 5). At low ice cover stations, Sva1033 increased close to surface sediment and peaked in abundance (>7%) at several centimeter depths where iron reduction predominates. Only sediments at the heavy ice cover station where the ferruginous zone is located at greater sediment depth, the relative abundance of *Sva1033* decreased (<2%) near to sediment surface while it increased (>4%) below 5 cm depth in the ferruginous zone (Figure 4). Members of this clade were detected previously in the ferruginous sediments of the (sub-) Antarctic [26] and Arctic shelf [25,58] and were suggested to be responsible for the iron reduction in these permanently cold marine sediments. Their metabolic capabilities to reduce iron have been confirmed recently by incubation experiments of Antarctic Potter Cove sediments using RNA stable isotope probing [26]. Our results provide further evidence that *Sva1033* plays an active role in the cycling of iron in Antarctic sediments.

Other putatively Fe-reducing bacterial taxa identified in this study besides Sva1033 (class *Desulfuromonadia*) are *SEEP-SRB4* (class *Desulfobulbia*), *Persicirhabdus* (class *Verrucomicrobiae*), *Halioglobus*, *order_B2M28* and *OM60 (NOR5)* clade (class *Gammaproteobacteria*), *Maribacter* and *Lutimonas* (class *Bacteroidia*) and class *Polyangia* (Figure 4 and Figure 5). Members of these groups make up the bulk of sedimentary anaerobic communities in the ferruginous zones, and some of them were shown previously to be present in marine sediments with high dissolved iron and manganese concentrations [52,70,71,72,80,96], suggesting an important role in meditating metal biogeochemical cycling. Moreover, the results of the environmental distribution for the most closely related sequences (>99% similarity) reveal that >70% of these taxa were detected before in permanently cold marine sediments (e.g., Antarctic and Arctic) [22,26,58,70,71,72,73,74] and various deep-sea environments [77,78,79,80,96], indicating that most of these taxa are psychrophilic (Figure 6).

An increase in the relative sequence abundance of putatively Fe-reducing bacterial taxa in the ferruginous zone at both high ice cover and low ice cover stations (Figure 4 and Figure 5) provides evidence that they might be involved in DIR in surface sediments of the AP or have syntrophic partnerships and/or common metabolic preferences with iron reducers. Future work should explore the in situ metabolic activity of these potential Fe-reducing bacterial taxa, with a particular focus on Sva1033, to better understand factors that control iron bioavailability and potential efflux into the water column from Antarctic and Southern Ocean shelf sediments.

The extent of Antarctic sea ice has undergone a drastic decline since 2015, indicating increased interannual variability [83]. Model projections further indicate a continued decline in the near future [97]. Our results show that the melting of sea ice sustains favorable light conditions and water column stratification, resulting in increased primary productivity and organic matter flux to the seafloor [32]. These changes, in turn, have an impact on the activity and composition of benthic microbial communities. Consequently, the regional (southward) shift in sea ice cover could potentially lead to an increase in benthic remineralization rates, promoting a shift in microbial communities towards anaerobic taxa and an increase in benthic iron fluxes. The latter could have positive feedback on primary production in the water column, further stimulating the overall process.

## 5. Conclusions

Our study reveals that sea ice cover and associated carbon burial fluxes explain up to 13% of the variability between microbial communities in the AP shelf sediments. At all stations, *Sva1033* was the dominant member of Desulfuromonadales in the ferruginous sediments, which confirms its putative role in reducing iron oxide minerals in permanently cold marine sediments. Furthermore, our approach identified for the first time other taxa that might contribute to the benthic iron cycle or have ecological relationships with the dominant iron reducers. The significant contribution of potential iron reducers (up to 15%) to microbial communities reveals the importance of sediments underlying low ice cover as a potential source of dissolved iron to shelf waters. In this regard, the findings reported here expand our knowledge about changes induced by the increase of OM load to microbial community composition in AP shelf sediments under contracting sea ice cover and stimulate future research to better elucidate the role of microbial iron reducers in the biogeochemical iron cycle.

## Figures and Tables

**Figure 2 microorganisms-11-01572-f002:**
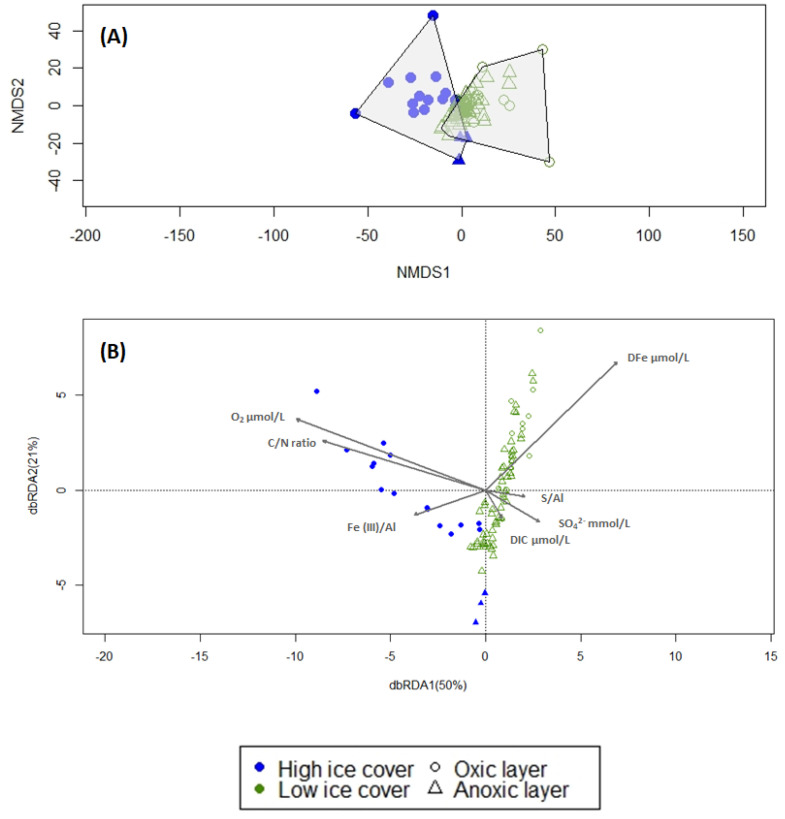
Microbial community composition of all five shelf stations. (**A**) Non-metric Multidimensional Scaling (NMDS) ordination of microbial community composition based on Euclidean distance after transforming the data using CLR for stations under high ice cover (Shelf St1) and low ice cover (Shelf St2, St3, St4, St7) (stress value = 0.152, R^2^ = 0.98). (**B**) Distance-based redundancy analysis (dbRDA) ordination plot of bacterial communities. Sample points in panels A and B are distinguished by site and sediment depth (oxic and anoxic layer) and by color and shape, respectively. dbRDA1 (variation 50%) and dbRDA2 (variation 21%) axes are displayed, which constrain the Euclidean distance matrix with geochemical parameters O_2_, DFe, DIC, SO_4_^2−^, Fe(III)/Al, S/Al, C/N ratio. The total model (F = 2.90, *p* < 0.001, Df: 7, 81) and each individual parameter (*p* < 0.05) were significant except for SO_4_^2−^.

**Figure 3 microorganisms-11-01572-f003:**
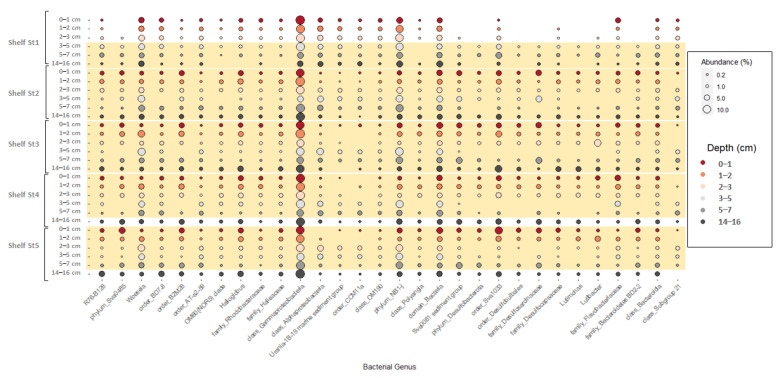
Microbial community composition in five shelf stations across the eastern coast of the Antarctic Peninsula. The relative abundance of bacterial 16S rRNA genes (cut-off > 0.1%) at genus-level resolution is shown for six sediment layers. For those taxa unclassified at genus, the higher taxonomic level is reported. Orange shadings represent the ferruginous zone in each station.

**Figure 4 microorganisms-11-01572-f004:**
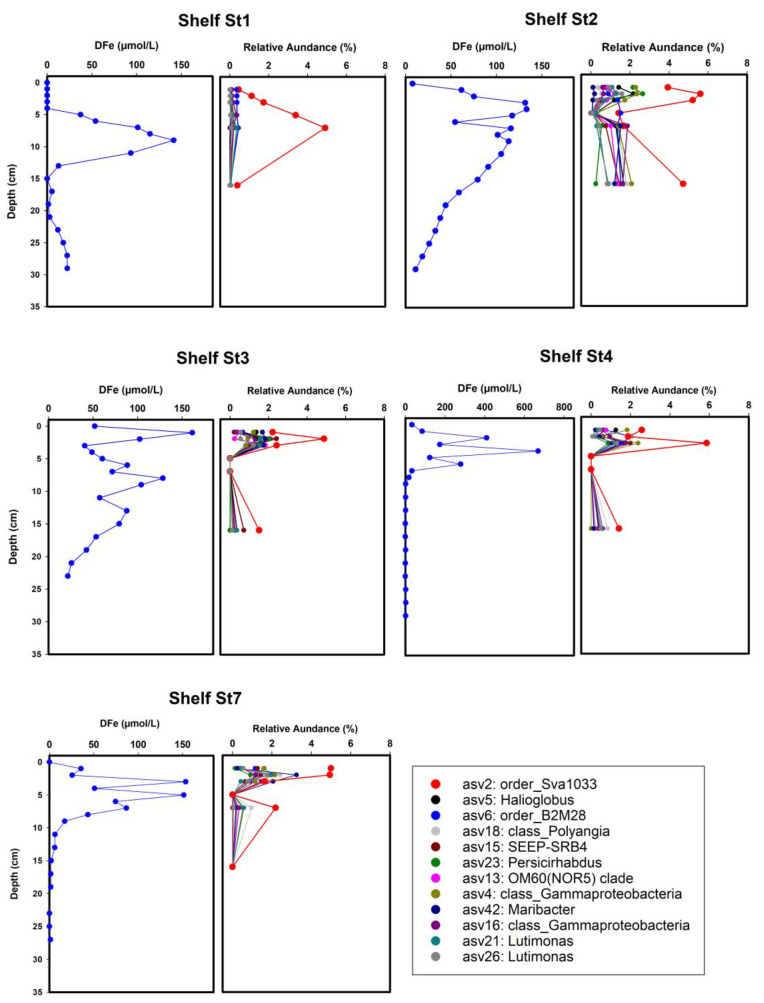
Representative profiles of the depth distribution of abundances of putative Fe-reducing bacteria in relation to dissolved iron (DFe) concentrations. Twelve different taxa were identified by applying differential abundance analysis. The relative abundance for each taxon was estimated from one sediment core. DFe concentrations were measured in a core near (<1 m) the collected microbial samples.

**Figure 5 microorganisms-11-01572-f005:**
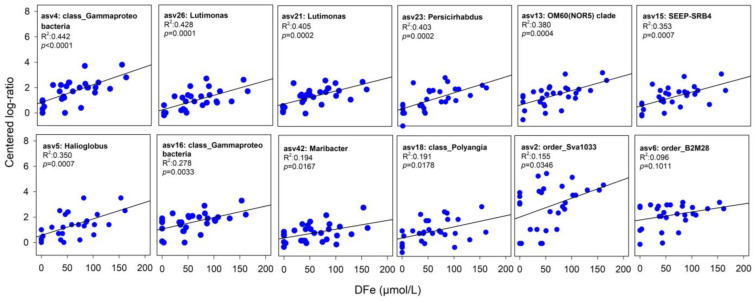
Linear regression between centered log-ratio values of putative Fe-reducing bacteria and DFe concentrations (in µmol/L) for five shelf stations. Linear models’ R^2^, Pearson’s correlations, and their *p*-value are reported in each panel. The centered log ratio of putative Fe-reducing bacteria used in the correlations is the average of two sediment cores. At Shelf St4, only five depths were used in the correlations. The last depth was excluded as it is below the ferruginous zone.

**Figure 6 microorganisms-11-01572-f006:**
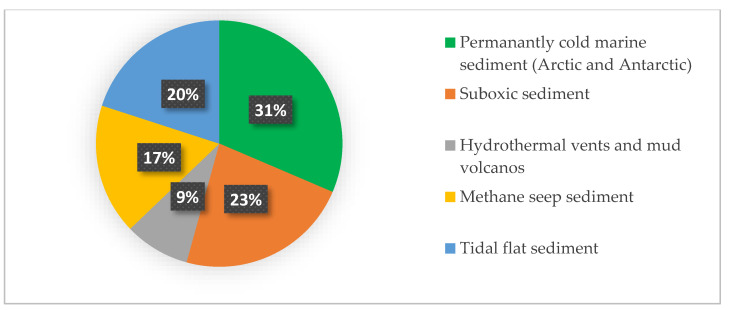
Environmental distribution for the most closely related sequences (>99% similarity) to ASV highly abundant in the ferruginous zone. For details, see Appendix A.

## Data Availability

Sequence data for this study have been deposited in the European Nucleotide Archive (ENA) at EMBL-EBI under accession number PRJEB57442 (https://www.ebi.ac.uk/ena/data/view/PRJEB57442), using the data brokerage service of the German Federation for Biological Data GFBio, (accessed on 9 June 2023) [99].

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
