# Peer review of "The Impact of Sea Ice Cover on Microbial Communities in Antarctic Shelf Sediments"

_microorganisms, 2023, doi:10.3390/microorganisms11061572_

Round 1
Reviewer 1 Report
The presented article is an original research material that is of scientific interest. It is devoted to the study of microbial communities living under various ice conditions in the bottom sediments of the Antarctic Peninsula.
It is devoted to the study of microbial communities living under various ice conditions in the bottom sediments of the Antarctic Peninsula.
the authors conducted a large-scale study, revealed the distribution of microbial communities depending on the redox situation and ice cover, respectively, linking the analysis with the input of organic matter.
The article is of interest, presents new knowledge and scientific data, I believe that the authors presented the material at a fairly high level, and the minor comments that I had should not interfere with the publication. I believe that the article can be published in this submission.
Author Response
Dear Editor,
We would like to thank the reviewer for the careful review. We are pleased that the feedback was positive, as the reviewer was satisfied with the motivation and results of our work.
Yours sincerely,
Marwa Baloza
Reviewer 2 Report
The article is well written, even though the manuscript is more descriptive than the heart-braking one, I like the results. I would like to see more information on the biota enclosed in the ice but still, the information is very valuable.
I recommend the article for publication.
Author Response
Response to Reviewer 2 Comments
Dear Editor,
We would like to thank the reviewer for the careful review. We are pleased that the feedback was positive, as the reviewer was satisfied with the motivation and results of our work.
Yours sincerely,
Marwa Baloza
Reviewer 3 Report
Microorganisms-2320056-peer-review_Round-1
General comments
This manuscript reports the relationship between the V3V4-based microbiomes and the ice cover-related biogeochemical profiles of shelf sediments off Antarctic Peninsula. Previous studies, e.g., Ref. [32], showed that high/low ice covers resulted in low flux of the photosynthetic primary production (PPP)-derived organic matter (OM) supply to the sediments, while moderate ice cover led to high OM supply.
Aerobic and anaerobic breakdown of OM is thus affected by sea ice covers, and the sediment microbiomes are to be affected, too: this is the authors’ hypothesis #1 (L353-353). According to their Conclusion, the hypothesis #1 was proved as “sea-ice cover and associated carbon burial fluxes explain up to 13% of the variability between microbial communities in the AP shelf sediments” (L486-487). This is a nice point of the study.
The authors’ hypothesis #2 (L353-354) would be interpreted as “iron reduction contributes to the anaerobic OM degradation, yielding Fe2+ or dissolved iron (DFe), by the action of iron-reducing bacteria that might dominate Fe(III)-rich ferruginous zones.” According to their Conclusion, the hypothesis #2 was tested as “significant contribution of potential iron reducers (up to 15%) to microbial communities reveals the importance of sediments underlying low ice cover as a potential source of dissolved iron to shelf waters” (L492-494), this is also a nice point but is rather speculative.
Anyway, these hypotheses should be presented earlier in Abstract and/or Introduction.
And, it may be an idea for the authors to add discussion about the influence of “climate change” that may affect the status of “sea ice cover” and eventually microbiomes and biogeochemical processes in the sediments.
Along with the Specific Comments, my overall evaluation of the manuscript is “Major Revision”.
Specific comments
L54-56 “Thus, the sea ice cover controls the flux of OM to the seafloor, and especially the occurrence of moderate sea ice cover (5-35%) was found to correlate with a high OM supply rate to the seabed [32].”
The authors may summarize the first several sentences of “4. Discussion” of Ref. [32], which would facilitate the readers’ understanding of somewhat complicated linkage between the “sea ice cover” and “flux of OM”; it is not so simplistic as “high ice cover, high OM supply” nor “low ice cover, high OM supply”.
L104 and L112 “a 400 mile transect”
Ref. [32] on which this study is based also uses the phrase “a 400 mile transect”. However, it is ambiguous in the main text whether the “mile” denotes the statute mile (international mile, 1609 m) or the nautical mile (1852 m). Although Figure 1 shows “Nautical Miles”, the figure caption uses a “km” instead of “mile”. Please state clearly “nautical mile” in the main text or prefer to use “km” consistently throughout the manuscript.
L112 and Figure 1 “sediments were collected from 5 stations along a 400 mile transect”
Ref. [32] states that “sediments were collected from 7 stations along a 400 mile transect”. The deep-sea stations 5 and 6 in Ref. [32] were not used in this “shelf” manuscript. The original station 7 in Ref. [32] corresponds to “St5” in this manuscript. It might be an idea to designate “St5” in this manuscript as “St7” to be consistent with Ref. [32] with clarification that the original stations 6 and 7 were not “shelf stations” and thus not dealt with in this study. By doing so, the comparison of data between Ref. [32] and this study, which provides an important perspective, will be easier. Please consider.
L165 “The rarefaction curves for each sample were generated…”
A major purpose of rarefaction is to estimate the “coverage” of the “observed richness” against “theoretical richness”. The asymptotes, or plateaus, of the rarefaction curves indicate the “theoretical richness” and correspond to the values of Chao1, an alpha-diversity index. The “coverage” is calculated by dividing the “observed richness” with “theoretical richness” (= Chao1 value). Therefore, Chao1 should be used as the major alpha-diversity index, rather than the exponential Shannon and inverse Simpson in Table S1.
Please note that singletons should be included in the rarefaction curves and Chao1 calculations.
L187-189 “Linear regression … was performed using Spearman’s rank correlations at p < 0.05.”
Please explain why the authors used the non-parametric “Spearman’s rank correlation” rather than the commonly used parametric correlations.
L190-192 “the environment coverage for … was identified with BLASTn … and reported for each ASV.”
Please clarify what “the environment coverage” means.
L196-197 “we obtained a final dataset of 4,252,065 reads (amplicons) in 90 samples, which were assigned to 38,655 ASVs belonging to 536 bacterial taxonomic families (Table S1).”
Please show the overall totals at the bottom of Table S1. Currently no information about the numbers of bacterial families is seen in Table S1. Not only the numbers of families but also the numbers of other taxonomic ranks should be shown in a separate table.
L197-199 “Rarefaction profiles of 16S rRNA gene sequences reached a plateau for most of the samples (Figure S1).”
Some samples did not reach plateaus, which is related to a serious problem of “sequencing depth” or “coverage” as mentioned above.
Table S1 shows that the smallest and largest numbers of the “amplicons after QC and merging” (singletons remained) were 3,718 and 612,485 of the samples St7_MUC1_C8_D4 and St3_MUC4_C2_D1, respectively, yielding about 165-fold difference.
The sequencing depths, or the numbers of screened amplicons, should be flattened among the samples by random-picking of amplicons and by discarding the samples that had too-small numbers of amplicons.
Without flattening, analyses of the datasets must have been inadequate.
Table 1
It is not bad to import the data directly from the Table 1 of Ref. [32]; however, the longitudes and latitudes should better be presented without “-” (minus) by replacing “E” and “N” with “W” and “S”, respectively.
L204-205 “Overall, the heavy ice-covered station had the highest species richness and diversity of all stations.”
If the “heavy ice-covered station” is “St1”, please state so clearly. Even though St1 had the heaviest ice cover %, the sea ice index of St1 was as low as 0.038, which was the lowest. Please explain the relationship between the ice cover % and the sea ice index and the meaning of these contrastive values at St.1.
L312-313 “Sva1033 …with a peak in the relative abundance at the same depth of the highest dissolved Fe concentrations”
The correlation in Figure 5 does not support this statement. If the authors assume that Sva1033 plays the major role in iron reduction, please explain the not-so-significant correlation in Figure 5. Realistically speaking, [Sva1033] and [DFe] may correlate both positively and negatively, as high [DFe] may mean the depletion of Fe(III) that would be the substrate for Sva1033 (and other iron-reducers).
Figure 5
Data (dots) from different stations should be presented in different colors.

Round 2
Reviewer 3 Report
The manuscript has been revised well and is now qualified for publication.